# The Impact of Urbanization on Tree Growth and Xylem Anatomical Characteristics

**DOI:** 10.3390/biology12111373

**Published:** 2023-10-27

**Authors:** Xiaohui Gao, Binqing Zhao, Zecheng Chen, Wenqi Song, Zongshan Li, Xiaochun Wang

**Affiliations:** 1Aulin College, Northeast Forestry University, Harbin 150040, China; 2Key Laboratory of Sustainable Forest Ecosystem Management-Ministry of Education, School of Forestry, Northeast Forestry University, Harbin 150040, China; 3State Key Laboratory of Urban and Regional Ecology, Research Center for Eco-Environmental Sciences, Chinese Academy of Sciences, Beijing 100085, China

**Keywords:** urbanization, urban heat island, air pollution, dendrochronology, tree ring anatomy

## Abstract

**Simple Summary:**

This study explored how urbanization (urban heat island effect and air pollution) affects urban tree growth. The radial growth and xylem anatomical characteristics of trees in urban and rural areas of Harbin were compared by means of tree-ring anatomy. Results showed that there were significant differences in the growth of both broadleaf trees and conifers between urban and rural areas. Urbanization may have the effect of slowing down tree growth, and conifers may be more sensitive to urbanization. Warming and drying climate in Harbin may be an important factor affecting tree growth. Our findings provide a theoretical basis for the future evaluation of the effect of urbanization on tree growth and ecological benefits, as well as the selection of tree species in urban forests.

**Abstract:**

In the context of the intensification of global urbanization, how urbanization (urban heat island effect and air pollution) affects urban tree growth is not fully understood. In this paper, the radial growth and xylem anatomical characteristics of three different tree species (*Quercus mongolica*, *Fraxinus mandshurica,* and *Pinus sylvestris* var. *mongolica*) in urban and rural areas of Harbin were compared by means of tree-ring anatomy. The results showed that there were significant differences in the growth of both broadleaf trees and conifers between urban and rural areas. The vessel number, cumulative area of vessels, and theoretical hydraulic conductivity of all tree species in rural areas were higher than those in urban areas, indicating that urbanization may have the effect of slowing down growth. However, broadleaf trees in urban areas had higher vessel density and a greater percentage of a conductive area within xylem and theoretical xylem-specific hydraulic conductivity. The thickness of cell walls and cell wall reinforcement index of *P. sylvestris* var. *mongolica* were strongly reduced by air pollution, implying that it may be more sensitive to urbanization. Compared to *Q. mongolica*, *F. mandshurica* showed less sensitivity to urbanization. Warming and drying climate in Harbin may be an important factor affecting tree growth.

## 1. Introduction

With the continuous increase of the world population, the pace of world urbanization will continue to accelerate. It is estimated that by 2050, the world urbanization rate will increase from 56% in 2021 to 68% [1]. Human activities in cities inevitably lead to heat and pollution gas emissions, resulting in two important consequences of urbanization: urban heat island effect and air pollution [2]. Urban heat island effect and air pollution have potentially large impacts on people and vegetation in cities and surrounding areas [3,4].

Urban trees are the most important part of urban vegetation. On the one hand, they reduce air pollution, absorb carbon dioxide, regulate air temperature and humidity, reduce storm-water runoff, etc. [5]. On the other hand, they are heavily affected by the process of urbanization, especially the urban heat island effect and air pollution [6]. Many scholars have conducted a lot of research on the improvement of the urban environment via urban trees [7,8,9,10]. However, there are relatively few studies on the effects of the urban heat island effect and urban pollution on urban trees [3], especially on the anatomy of their trunk xylem. Studying the impact of urbanization on tree anatomy can help us observe changes in the impact of urbanization on tree growth from a more microscopic perspective. Due to the impact of urbanization, there are many differences in the environment between urban areas and surrounding natural areas, such as temperature, precipitation, and air pollution, which lead to different living conditions of trees [6]. High temperature and drought from the urban heat island effect may cause trees to create smaller vessels, while air pollution may prevent trees from forming vessels [11,12]. Therefore, the environmental differences between urban areas and surrounding natural areas and the impact of urbanization on urban vegetation growth have attracted more and more scholars’ attention [4].

Urban heat island (UHI) effect is the result of the physical properties of buildings and other structures, as well as the heat emitted by human activities [13]. The average UHI trend for global cities can reach 2.36 ± 0.69 °C at night and 1.60 ± 1.50 °C during the day [14]. UHI results in temperature differences between urban and rural environments, with consequences for urban vegetation, including altered plant phenology, extended growing seasons, and increased susceptibility to drought stress and insect infestation [2,15,16,17,18,19,20]. At the same time, many studies have proved that there is a correlation between the heat island effect and drought in urban areas [21,22,23,24,25,26]. Trees in cities respond to the urban heat island effect [27]. For example, Moser-Reischl et al. found urban tree growth is usually negatively affected by urban growing conditions [28]. Urban areas can create stressful environments for native trees, leading to increased mortality and ecosystem changes [29]. Rising temperatures and more frequent heat and drought events will exacerbate stress levels in urban trees [30]. The long-term impact of rising temperatures will reduce productivity, as warmer weather and slower tree growth exacerbate air pollution and seasonal climate change [31]. Compared to trees in rural environments, trees from urban areas are more sensitive and vulnerable to the climate [32,33].

Many studies have suggested that urbanization has a negative impact on tree growth. However, some studies also found that urbanization has a positive indirect impact on tree growth, although the land use change brought about by urbanization led to a direct reduction in the number of vegetation [4]. Some urban trees experienced accelerated growth in the past decades [6,34,35,36]. Urban trees take advantage of warm conditions despite the urban heat island effect and moisture limitations [5]. Other studies found no differences in radial growth dynamics between urban and rural trees, except for extreme climatic events in individual years, although the climate conditions in urban areas are warmer [37,38]. However, these studies basically focus on the study of radial growth and isotopes of trees and do not explain and discuss from the microscopic perspective of xylem anatomy.

Air pollutants refer to substances that, at certain concentrations in the atmosphere, may harm the health of organisms, affecting regional air quality and global climate [2,39]. The most frequently monitored air pollutants in urban areas include particulate matter (PM_10_, PM_2.5_, UFP), NO_2_, CO, SO_2_, and O_3_. Many studies suggest that air pollution negatively affects tree growth. Thus, tree rings can be used as archives of past air pollution [27,40,41,42,43,44,45]. The mechanism of air pollution affecting tree growth has been extensively studied [3,42,46,47,48]. The emission of SO_2_, NO_x_, CO_2_, dust, and other pollutants seriously interferes with the metabolism of trees [27]. Particulate matter can be adsorbed on leaf surface or cuticle of trees, or it can be metabolized by trees through the stomata or rhizosphere [48,49]. These changes will ultimately affect the photosynthetic rate and transpiration of trees [50,51]. Compared with urban heat island effect, air pollution directly affects tree growth, which is a major growth-limiting factor [5]. Air pollution increases the sensitivity of tree growth to climate factors [44]. Therefore, urban trees exposed to pollution sources are more vulnerable to extreme climate change [52]. Likewise, these studies only quantified tree growth through the radial width but did not explore the impact of pollution effects on tree growth from a more microscopic perspective.

Harbin is the provincial capital of Heilongjiang Province, the largest city in Northern China and an important manufacturing base [53]. In recent years, the urbanization process of Harbin has been very fast, and the environmental problems of the city have also become prominent, e.g., the urban heat island effect and air pollution caused by industrial emissions, road traffic, straw burning, and centralized heating in winter [54,55,56,57]. *Quercus mongolica*, *Fraxinus mandshurica,* and *Pinus sylvestris* var. *mongolica* are the main afforestation tree species in Harbin, and the amount of *P. sylvestris* var. *mongolica* and *Q. mongolica* tends to increase continuously. At the same time, as a high-latitude city in the northern hemisphere, Harbin is also one of the cities with the most significant temperature increases. However, whether urban heat island effect and air pollution in Harbin will aggravate the impact on tree growth in the context of climate warming is unclear. Therefore, related research is urgently needed.

This study aimed to investigate the impact of urbanization on radial growth and xylem anatomical characteristics of trees so as to provide a theoretical basis for evaluating the impact of urbanization on tree growth and ecological benefits and the selection of urban forest tree species in the future. To this end, we propose the following hypothesis: (1) There are differences in growth between urban and rural trees in Harbin, and the growth of trees in rural areas is better than that in urban areas (Harbin); (2) Air pollution will reduce tree growth, but urban heat island effect may promote tree growth; (3) Urban trees will have more and smaller vessels than rural trees. We temporarily ignore differences in soil, elevation, and natural and plantation forests in this study.

## 2. Materials and Methods

### 2.1. Study Areas

In order to study the impact of urban heat island effect and air pollution on tree growth, we selected Harbin as the urban sampling areas (HRB), and Maoershan Forest Ecosystem National Field Scientific Observation and Research Station (LM) 90 km away from Harbin as the control sampling site (rural areas). It was divided into two gradients of air pollution in Harbin. The lightly polluted site is located in the Northeast Forestry University Urban Forestry Demonstration Base, Harbin, China (45°43′ N, 126°37′ E), and another site is on the side of Shuini Road, Daowai District, Harbin, China (45°50′ N, 126°43′ E), which is more polluted. Tree-ring sampling of *Q. mongolica*, *F. mandshurica,* and *P. sylvestris* var. *mongolica* was conducted at the above sites, respectively (Figure 1, Table A1).

The climate belongs to the mid-temperate continental monsoon climate, with long winters and short summers. The mean annual total precipitation is 569.1 mm. The mean annual temperature is 3.6 °C. The frost-free period is 140 d. The average elevation of Harbin is 151 m. The soil type of Harbin Experimental Forest Farm is black soil [58]. The pollutants discharged are mainly SO_2_, NO_x_, and smoke (powder) dust [59].

The main forest type of the Urban Forestry Demonstration Base of Northeast Forestry University (NF) is the artificial pure forest planted from the late 1950s to the early 1960s. There are about 30 pure artificial forests in the sampling areas, and the main tree species include *Q. mongolica*, *P. sylvestris* var. *mongolica*, *Betula platyphylla*, *F. mandshurica*, *Juglans mandshurica*, etc. [60]. Except for *Q. mongolica* oak forest, a large number of shrubs and herbs have grown under the plantations. The main shrubs include *Sambucus williamsii*, *Ligustrum obtusifolium* subsp. *suave*, *Caragana arborescens*, *Syringa reticulata* subsp. *Amurensis*, and *Syringa oblata*, and the herbs include *Chelidonium majus*, *Hosta plantaginea*, and *Trigonotis peduncularis*.

Shuini Road (SR), with an altitude of ~132–140 m, is near the Harbin Cement Company of Yatai Group. The sampling site was set in a park about 100 m away from the factory building. The park covered an area of about 30,000 square meters. The soil type is black-brown. The main tree species planted in the park were *Salix matsudana* and *P. sylvestris* var. *mongolica. Sambucus williamsii*, *Lespedeza bicolor*, *Pilea pumila*, *Amphicarpaea edgeworthii,* and *Ixeris polycephala* distribute under the forest.

The sampling site of the forest areas in contrast to Harbin is located at the National Field Scientific Observation and Research Station of Maoershan Forest Ecosystem in Heilongjiang Province (LM), Shangzhi, China (45°25′ N, 127°38′ E). It is 90 km southeast of Harbin. This area belongs to the temperate continental monsoon climate. The annual mean temperature is 2.8 °C. The frost-free period is between 120–140 d. The mean annual total precipitation is 723 mm, and the mean annual evaporation is 1093 mm. The zonal soil is dark brown forest soil.

The National Field Scientific Observation and Research Station of Maoershan Forest Ecosystem, Maoershan belongs to the branch of Zhangguangcai ridge, with an average altitude of 380 m. The main vegetation is the natural secondary forest formed by the broadleaf Korean pine forest after different degrees of disturbance (harvesting, management, burning, reclamation, etc.). The main tree species are *Pinus koraiensis*, *F. mandshurica*, *Juglans mandshurica*, *Q. mongolica*, *Tilia amurensis*, *Betula platyphylla*, etc.

The climate data were aggregated for both research areas. In general, the average annual temperature of HRB was 1.3 °C higher than that of LM, while the precipitation was 124.2 mm less than that of LM (Figure 2). From a yearly perspective, HRB had a higher temperature, maximum temperature, minimum temperature, and vapor pressure deficit (VPD), as well as lower precipitation (Figure A1). From 2014 to 2020, the Air Quality Index (AQI) of HRB was decreasing, indicating that air pollution is gradually improving. In general, HRB is warmer and drier than LM.

### 2.2. Sampled Tree Species

The sampled tree species of this research are *Q. mongolica*, *F. mandshurica,* and *P. sylvestris* var. *mongolica*. *Q. mongolica* (qm) is a widely distributed tree species in the broadleaf forests of Northeast China and is susceptible to aggravated drought [61]. It has great value as a highly exploited forest resource and is also included in the list of 100 forest plant species vulnerable to climate change [62]. *F. mandshurica* (fm) is one of the most important temperate hardwood species in the mixed broadleaf—Korean pine forest in Northeast China. It is highly sensitive to climate change and is extensively distributed but discontinuous in Northeast China, Eastern Russia, Northern Japan, and North Korea [63]. *P. sylvestris* var. *mongolica* (ps) is the most widely distributed evergreen conifer in the northeastern part of boreal Asian forests [64]. It is also one of the main test objects in dendrochronology research due its role in habitat formation, strong adaptability, distinct annual rings, and high sensitivity [42].

### 2.3. Sampling and Chronology Development

In 2021, we collected tree ring cores at three sites. Species sampled included two broadleaf trees (*Q. mongolica*, *F. mandshurica*) and one conifer (*P. sylvestris* var. *mongolica*). Smaller trees, such as those with DBH < 15 cm, were not collected because they might suffer from more competition or other ecological factors. Two or three cores were collected at breast height (~1.3 m) from healthy living individuals with an increment borer (inner diameter: 5.15 mm; Haglöf, Dalarna, Sweden). A total of 321 cores from 140 trees were collected in three sites. Table 1 shows the cores used in developing the chronology; an additional five cores from each tree species at each site were included in the anatomical analysis. The cores were fixed, air-dried, and sanded gradually until each ring was clearly visible. Each ring was then cross-dated to identify possible deletions or false rings, and an accurate calendar year was assigned using the skeleton-plot method under the microscope [65]. Subsequently, the width of the rings was measured with a Velmex measuring system (UniSlide A40 Series, New York, NY, USA) with an accuracy of 0.001 mm. The accuracy of the results was checked using the COFECHA program [66].

The chronologies were developed and the non-climatic effects on tree growth were removed using a modified negative exponential curve (ModNegExp) detrending method by dplR package [67]. The length, standard deviation (SD), mean sensitivity (MS), first-order autocorrelation (AC1), mean inter-series correlation (Rbar), signal-to-noise ratio (SNR), expressed population signal statistic (EPS) [68], and average growth rate (AGR) of the tree-ring chronologies were calculated using dplR package in the R statistical environment [67]. ANOVA was performed to observe the differences in growth between urban and rural trees. Results were tested using the least-significant difference method by agricolae package [69]. Correlation histograms were plotted using the Spearman correlation analysis method by linkET package [70] to analyze the correlation coefficients between monthly values, climatic factors, and radial growth. All analyses and plots were made with R statistical software (R version 4.2.2) [71].

### 2.4. Anatomical Characteristics Analysis

The tree ring cores were softened in an 89-degree water bath and preprocessed with a GSL1–microtome [72]. The sections were stained with safranin reagent, rested for 5 min, washed with water and 75% ethanol, and finally mounted with 33% glycerin [73]. A semiautomatic digital camera (E3ISPM05000KPA, Zhengzhou, China) connected to a microscope (XSP–GX6A, Shanghai, China) was used to convert the clear slice into a digitized image at 10× magnification with a spatial resolution of 1.52 pixels/µm. Subsequently, professional image analysis software such as Image–Pro Plus (version 6.1) and ROXAS (version 3.0) were used to automatically identify tree-ring image data and measure xylem anatomical characteristics [74].

Tree-ring anatomical parameters, including the mean ring width (MRW, μm), vessel number (VN), mean vessel density (VD, No.∙mm^−2^), cumulative area of all counted vessels (CTA, mm^2^), mean percentage of conductive area within xylem (RCTA, %), mean vessel area per rings (MVA, μm^2^), theoretical hydraulic conductivity (Kh, m^4^∙MPa^−1^∙s^−1^), theoretical xylem-specific hydraulic conductivity (Ks, m^2^∙MPa^−1^∙s^−1^), and overall mean hydraulic diameter (Dh, μm) of broadleaf trees were measured and calculated. For conifers, we calculated the mean ring width (MRW, μm), cell (tracheid) number (CN), mean cell density (CD, No.∙mm^−2^), cumulative area of all counted cells (CTA, mm^2^), mean percentage of conductive area within xylem (RCTA, %), mean cell lumen area per rings (MLA, μm^2^), theoretical hydraulic conductivity (Kh, m^4^∙MPa^−1^∙s^−1^), theoretical xylem-specific hydraulic conductivity (Ks, m^2^∙MPa^−1^∙s^−1^), overall mean hydraulic diameter (Dh, μm), overall mean thickness of all cell walls (CWT, μm), and overall mean cell wall reinforcement index (TB2) [75].

The one-way ANOVA and two-way ANOVA analysis of different tree species and sites were performed for each anatomical parameter to observe the differences in growth between urban and rural trees. Results were tested using the least significant difference method by agricolae package [69]. To evaluate which type of anatomical parameters were most influenced by climate change in urban and rural areas, we used principal component analysis to reduce dimensionality and find the dominant factor characteristics of the first and second principal components and analyze their correlation with climatic factors, respectively. Package factoextra and FactoMineR were used in this step [76,77]. Correlation histograms and heat maps were plotted using the Spearman correlation analysis method by linkET package [70] to analyze the correlation coefficients between monthly values, climatic factors, and anatomical parameters, as well as which factors might mainly affect tree ring parameters. All analyses and plots were made with R statistical software [71].

### 2.5. Climate Data

The climate data were from the National Meteorological Science Data Center (China Meteorological Data Network, http://data.cma.cn/, access on 31 December 2020). Climate variables include air temperature (TEM), precipitation (Prec), ground temperature (GTEM), maximum air temperature (T_max_), minimum air temperature (T_min_), and relative humidity (RH). Standardized Precipitation Evapotranspiration Index (SPEI) was calculated using the SPEI package [78], and Vapor Pressure Deficit (VPD) was calculated by Equation (1).
(1)$$VPD=0.61078\times e^{\frac{17.27\times TEM}{TEM+237.3}}\times (1-RH)$$

Two sampling sites (NF and SR) shared the same climate data of Harbin Meteorological Station, and the climate data of LM site were from Shangzhi Meteorological Station. Air pollution data were obtained from China Air Quality Online Monitoring and Analysis Platform (https://www.aqistudy.cn/, access on 28 February 2022) and Harbin Ecological Environment Bureau (http://www.harbin.gov.cn/, access on 31 January 2022). The air quality index data of LM was from Shangzhi Monitoring Station. Since Shangzhi is a city, it will be slightly higher than that in the Maoershan forest areas. Variables include Air Quality Index (AQI), PM_2.5_, PM_10_, NO_2_, CO, SO_2_, and O_3_–8 h.

## 3. Results

### 3.1. Radial Growth of Trees in Urban and Rural Areas

Tree-ring chronology lengths ranged from 39 years to 77 years (Figure 3 and Table 2). *F. mandshurica* and *Q. mongolica* from rural areas are natural forests with an average age of more than 70 years, while those from urban areas are plantations with an average age of 50–60 years. The average age of *P. sylvestris* var. *mongolica* from heavily polluted areas is even shorter at just under 40 years. The results of the statistical characteristics of the chronologies showed that there was a good mean inter-series correlation (Rbar) between most of the tree samples, indicating a consistency in the variation of ring width between different cores. The first-order autocorrelation coefficient (AC1) of the chronologies fluctuated widely, indicating that the growth status of different tree species in the previous year had a different impact on the current year. The mean sensitivity (MS) of each chronology was 0.28, indicating that the sample had a strong signal of high-frequency fluctuations. Except for *P. sylvestris* var. *mongolica* in heavily polluted areas, the signal-to-noise ratio (SNR) of the chronologies was high, and the expressed population signal statistics (EPS) reached above 0.85, indicating that the signal contained in the sample size collected in this study was basically representative of the overall characteristics [79].

The results of the ANOVA showed that the ring width of both *F. mandshurica* and *Q. mongolica* in urban areas were significantly lower than those in rural areas, but there was no significant difference in the ring width of *P. sylvestris* var. *mongolica* (Figure 4). When the chronology from 1974 to 2020 was segmented, it was found that there was no significant difference between urban and rural *F. mandshurica* from 1974 to 1989, but there was a significant difference between them from 1990 to 2020 (Figure A2A,D,G). Similarly, the differences between urban and rural *Q. mongolica* were progressively significant from 1974 to 2020 (Figure A2B,E,H). Both broadleaf trees had significantly higher ring width in rural areas than in urban areas, but the situation was very different for conifers. From 1974 to 1989, there was a significant difference between urban and rural *P. sylvestris* var. *mongolica*, but this difference disappeared from 1990 to 2004 (Figure A2C,F). From 2005 to 2020, the ring width of *P. sylvestris* var. *mongolica* in rural areas was again significantly higher than in urban areas, but the ring width in heavily polluted areas was higher than that in lightly polluted areas (Figure A2I).

The chronology of *F. mandshurica* was overall positively correlated with monthly SPEI, with a higher but non-significant coefficient during the growing season (Figure 5A). The chronology of urban *Q. mongolica* was positively correlated with monthly SPEI and had a significant positive correlation with monthly SPEI during the growing season, while the chronology of rural *Q. mongolica* had a non-significant positive correlation with monthly SPEI only during the growing season (Figure 5B). The chronology of *P. sylvestris* var. *mongolica* in lightly polluted areas was significantly and positively correlated with monthly SPEI, while the chronology of *P. sylvestris* var. *mongolica* in rural areas was significantly and negatively correlated with monthly SPEI (Figure 5C). The chronology of *P. sylvestris* var. *mongolica* in heavily polluted areas, on the other hand, was positively correlated with monthly SPEI during the year before the end of the growing season, with a higher but non-significant coefficient during the growing season (Figure 5C).

### 3.2. Xylem Anatomical Characteristics of Trees in Urban and Rural Areas

For *F. mandshurica*, VD, RCTA, and Ks in urban areas were significantly higher than those in rural areas, while VN, CTA, and Kh in rural areas were significantly higher than those in urban areas (Figure 6). Other parameters had no significant differences between urban and rural sites. For *Q. mongolica*, VD, RCTA, MVA, and Ks in urban areas were significantly higher than those in rural areas, while VN, CTA, Kh, and Dh in rural areas were significantly higher than those in urban areas. The common points of parameters of *F. mandshurica* and *Q. mongolica* in comparison between urban and rural areas were VD, RCTA, Ks, VN, CTA, and Kh. The difference was that *Q. mongolica* had significant differences in MVA and Dh, while *F. mandshurica* did not. The results showed that there were significant differences in most xylem anatomical parameters between urban and rural trees (Figure 6).

The two-way ANOVA of the effect of site (urban and rural) and species (*Q. mongolica* and *F. mandshurica*) on the anatomical parameters of xylem of trees showed that the effects of sites on most parameters were significant (Table 3). Except Ks and RCTA, species also had significant effects on anatomical parameters. Similarly, the interaction between sites and species had important influence on anatomical parameters, as most of the parameters were significant except Ks (Table 3).

It can be seen from the comparison of areas with different urban pollution levels (heavy pollution—SR, and light pollution—LM) that air pollution has led to a significant decline in CN, CTA, and Kh of *P. sylvestris* var. *mongolica*, while Dh in urban areas was significantly higher than that in rural areas (Figure 7). In the two pollution concentration gradients in Harbin, the differences between CWT, TB2, and Kh were more significant than those between urban and rural areas (Figure 7).

### 3.3. PCA Dimensionality Reduction

The principal component analysis of the main xylem anatomical parameters of the three species in urban and rural areas showed that (Figure 8 and Figure 9) the first component (PC1) of *F. mandshurica* in urban areas (46.4%) was mainly composed of Kh, MVA, CTA, and Dh (Figure 8A), while the PC1 in rural areas (51.5%) was mainly composed of Ks, Kh, RCTA, and CTA (Figure 8B). The second component (PC2) of *F. mandshurica* in urban areas (30.4%) was mainly composed of RCTA, VD, and Ks (Figure 8A), while the PC2 in rural areas (26.1%) was mainly composed of VN (Figure 8B). Compared with urban and rural areas, the main anatomical parameters exchanged in PC1 and PC2, and their contributions, were completely opposite (Figure 8A,B).

The PC1 of *Q. mongolica* in urban areas (52.8%) was mainly composed of VN, CTA, Kh, and VD (Figure 8C), while the PC1 in rural areas (43.7%) was mainly composed of Ks, MVA, RCTA, and Kh (Figure 8D). The PC2 of *Q. mongolica* in urban areas (27.9%) was mainly composed of Ks and RCTA (Figure 8C), while the PC2 in rural areas (25.3%) was mainly composed of VN (Figure 8D). Like *F. mandshurica*, compared with the urban and rural areas, the main anatomical parameters exchanged in PC1 and PC2, and their contributions, were completely opposite (Figure 8C,D).

The contribution of each anatomical parameter to PC1 and PC2 of *P. sylvestris* var. *mongolica* was similar at both urban pollution levels (Figure 9A,B). The PC1 of *P. sylvestris* var. *mongolica* in two urban sites (62.1% and 63.1%) were mainly composed of MLA, Ks, RCTA, Dh, and CWT (Figure 9A,B), while the PC1 in the rural site (56.3%) was mainly composed of MLA, Kh, RCTA, CTA, CD, and TB2 (Figure 9C). The PC2 of *P. sylvestris* var. *mongolica* in two sites (30.9% and 20.5%) were mainly composed of CTA and CN (Figure 9A,B), while the PC2 in rural areas (28.9%) was mainly composed of Dh, CWT, and Ks (Figure 9C). Similar to *F. mandshurica* and *Q. mongolica*, the contribution of anatomical parameters of the xylem of *P. sylvestris* var. *mongolica* to the principal component axis is opposite in urban and rural areas (Figure 9A–C).

### 3.4. The Relationship between Xylem Anatomical Characteristics of Trees and Climate

The PC1 (Kh, MVA, CTA, Dh) of the main anatomical parameters of *F. mandshurica* xylem in urban mainly represented hydraulic efficiency and safety (Figure 8A). The PC1 of *F. mandshurica* xylem in urban areas was not significantly correlated with monthly SPEI (Figure 10A). The PC1 (Ks, Kh, RCTA, CTA) of the main anatomical parameters of *F. mandshurica* xylem in rural areas mainly represented hydraulic efficiency and safety (Figure 8B). The PC1 of *F. mandshurica* xylem in rural areas was positively correlated with monthly SPEI, and the non-growing season coefficient was higher but not significant (Figure 10A). The PC2 of the main anatomical parameters of *F. mandshurica* xylem in urban and rural areas mainly represented vessel growth (Figure 8A,B). The relationship between PC2 of the main anatomical parameters of *F. mandshurica* and monthly SPEI was similar to that of PC1. That is, the PC2 of *F. mandshurica* xylem in rural was positively correlated with monthly SPEI (Figure 10B). Similarly, the non-growing season coefficient was higher but not significant. The PC2 of *F. mandshurica* xylem in urban was significantly positively correlated with monthly SPEI at the end of the growing season (Figure 10B).

The relationship between the principal components of the main xylem parameters of *Q. mongolica* in rural and monthly SPEI was quite different from that of *F. mandshurica* (Figure 11A,B). The PC1 (VN, CTA, Kh, VD) of the main anatomical parameters of *Q. mongolica* xylem in urban areas mainly represented vessel growth (Figure 8C). The PC1 of *Q. mongolica* xylem in urban areas was positively correlated with monthly SPEI, and the autumn coefficient was higher but not significant (Figure 11A). The PC1 (Ks, MVA, RCTA, Kh) of the main anatomical parameters of *Q. mongolica* xylem in rural areas mainly represented hydraulic efficiency and safety (Figure 8D). The relationship between PC1 of the main anatomical parameters of *Q. mongolica* and monthly SPEI in rural areas was not significant (Figure 11A). The PC2 (Ks and RCTA) of the main anatomical parameters of *Q. mongolica* xylem in urban areas mainly represented hydraulic efficiency and safety (Figure 8C). The PC2 of *Q. mongolica* xylem in urban areas was not significantly correlated with monthly SPEI (Figure 11B). The PC2 (VN, Dh, CTA, and Kh) of the main anatomical parameters of *Q. mongolica* xylem in rural areas mainly represented vessel growth (Figure 8D). The PC2 of *Q. mongolica* xylem in rural areas was positively correlated with monthly SPEI and significant in non-growing season (Figure 11B).

The relationship between the principal components of main xylem parameters of *P. sylvestris* var. *mongolica* and monthly SPEI was similar between PC1 and PC2 (Figure 12A,B). The PC1 (MLA, Ks, RCTA, Dh, and CWT) of the main anatomical parameters of *P. sylvestris* var. *mongolica* xylem in urban areas mainly represented hydraulic efficiency and safety (Figure 9A,B). The PC1 of *P. sylvestris* var. *mongolica* xylem in urban was positively correlated with monthly SPEI, and the autumn coefficient was higher but not significant (Figure 12A). The PC1 (MLA, Kh, RCTA, CTA, and TB2) of the main anatomical parameters of *Q. mongolica* xylem in rural areas mainly represented hydraulic safety (Figure 9C). The PC1 of *P. sylvestris* var. *mongolica* xylem in rural areas was negatively correlated with monthly SPEI, and the spring coefficient was significant (Figure 12A). The PC2 (CTA and CN) of the main anatomical parameters of *P. sylvestris* var. *mongolica* xylem in urban areas mainly represented hydraulic safety (Figure 9A,B). The PC2 of *P. sylvestris* var. *mongolica* xylem in urban areas was not significantly correlated with the SPEI (Figure 12B). The PC2 (Dh, CWT, and Ks) of the main anatomical parameters of *Q. mongolica* xylem in rural areas mainly represented hydraulic efficiency (Figure 9C). The PC2 of *P. sylvestris* var. *mongolica* xylem in rural areas was negatively correlated with monthly SPEI but was not significant (Figure 12B).

In general, the PC2 of the main anatomical parameters of *Q. mongolica* xyelm and the main anatomical parameters of *P. sylvestris* var. *mongolica* xylem in rural areas were more sensitive to SPEI, but *Q. mongolica* was mostly positively correlated, while *P. sylvestris* var. *mongolica* was mostly negatively correlated (Figure 11B and Figure 12A).

Except MVA, the anatomical parameters of *Q. mongolica* in rural areas were mainly negatively correlated with monthly SPEI, while the anatomical parameters of *Q. mongolica* in urban areas were mainly positively correlated with monthly SPEI (Figure A3). However, the correlation between *F. mandshurica* and monthly SPEI was much weaker; only Dh in rural areas and VD in urban areas were significantly positively correlated in the last months of the year (Figure A3). The responses of *P. sylvestris* var. *mongolica* to SPEI can be divided into two categories: growth anatomical parameters (CN, CD, CWT, and TB2) were generally positively correlated, while hydraulic anatomical parameters (CTA, RCTA, MLA, Kh, Ks, and Dh) were generally negatively correlated (Figure A3). Their response months were also different. The responses of *P. sylvestris* var. *mongolica* in urban areas to SPEI were concentrated in the second half of the year, while the responses of *P. sylvestris* var. *mongolica* in rural areas were distributed from the previous growing season to the first half of the year (Figure 12 and Figure A3).

## 4. Discussion

The study found significant differences in radial growth and xylem anatomical characteristics between urban and rural populations in both broadleaf trees and conifers. The changes of VN, CTA, and Kh were basically stable in all tree species, i.e., higher in rural areas than in urban areas (Figure 6). The difference of *Q. mongolica* between urban and rural areas was quite large, while *F. mandshurica* did not respond strongly to climatic factors. *P. sylvestris* var. *mongolica* may be more strongly affected by urbanization than the two broadleaf tree species.

### 4.1. Effect of Urban Heat Island on Anatomical Characteristics of Trunk Xylem

The urban heat island effect seems to have a negative impact on the growth of trees in Harbin. However, it has been debated whether the heat island effect promotes or hinders tree growth. Some studies have shown that long-term temperature increases in urban forests ultimately reduce productivity and slow tree growth [31]. Urban trees are sensitive to high temperatures and drought, which is thought to be one of the main factors limiting radial tree growth [33]. However, quite a few researchers have found that the urban heat island effect promotes tree growth. It has been noted that urban trees have experienced accelerated growth since the 1960s and tend to grow faster than trees in rural areas because urban trees are able to take advantage of warm conditions without being limited by water availability [5,35,36]. Street trees benefit from high temperatures and low precipitation in built-up urban areas [6]. It was also found that despite warmer climatic conditions in urban areas, there was no difference in the growth of urban and rural trees [37,38]. Affected by the heat island effect, the urban environment of Harbin is warmer and drier than the surrounding rural environment (Figure A1). For urban trees, warming and drying in Harbin decreased xylem anatomical parameters such as VN, CTA, and Kh. However, VD, RCTA, and Ks of broadleaf trees and Dh of conifers seem to be promoted by the urban heat island effect, which may be able to compensate for the decrease in hydraulic conductivity caused by the reduction in the number and total areas of vessels. We found that the growth decline of *P. sylvestris* var. *mongolica* in Harbin appeared to be more serious at the end of the 20th century and the beginning of the 21st century (Figure 3), which was considered to be the period when the urban heat island effect was strongest [54]. However, broadleaf trees did not show a similar trend, which may be due to the compensating mechanism mentioned above. In response to the seasons, Gillner et al. (2014) found that temperature and moisture availability from April to July of the current year and the summer and autumn of the previous year were the main determinants of radial growth. Broadleaf tree species were more sensitive to summer heat and drought in urban areas (summer signal), while conifers were suitable for analyzing urban heat islands in late winter and early spring (winter signal) [33]. For radial growth, our study was broadly consistent with this conclusion but not for anatomical parameters, implying that radial growth and anatomical characteristics did not correspond exactly.

Our results showed that the urban heat island effect in Harbin led to a decrease in the vessel area of all tree species (Figure 6C and Figure 7E). Islam et al. found that diffuse porous *Chukrasia tabularis* decreased vessel area under the stress environment, particularly in hot and dry conditions [80]. *E. grandis* seedlings grown at higher temperatures had a 40% reduction in the lumen area of xylem vessels [81]. At the same time, the urban heat island effect in Harbin resulted in increased vessel density of broadleaf tree species (Figure 6B). Islam et al. found that, under hot and dry conditions, not only the vessel area (hydraulic efficiency) decreased, but also the vessel density (hydraulic safety) increased [80]. Gea-Izquierdo et al. showed that trees adapted to more xeric conditions by reducing radial growth and hydraulic diameters, as well as increasing vessel density, but xylem was not adapted to respond to long-term temperature increases [82]. Under heat treatment, *Salix pulchra* Cham. had higher vessel density than ambient-growing plants [83]. However, some researchers have published contrary conclusions. Subfossil oaks from the Younger Dryas cold period, up to 3.5 °C cooler than present temperatures, had more vessel frequencies than living oaks [84]. In Central Europe, the vessel density of pedunculate oak (*Quercus robur* L.) was negatively correlated with the temperature of the current year in floodplain sites [85]. In Harbin, in order to adapt to a drier, warmer urban environment, trees sought a trade-off between maximizing conductivity and creating safety margins against cavitation and embolism, resulting in smaller vessel area and greater vessel density [80].

The urban heat island effect in Harbin also significantly increased the theoretical xylem-specific hydraulic conductivity (Ks) of both broadleaf tree species, while that of conifer species was not significantly increased (Figure 6G and Figure 7I). Similarly, the specific xylem conductivity of *Salix pulchra* Cham. after heat treatment was 2.5 times higher than that of the ambient-growing plants [83]. For ponderosa pine, larger tracheid diameters at elevated temperatures caused an increase in xylem-specific hydraulic conductivity [86]. However, Voelker et al. (2012) found that Younger Dryas cold period subfossil oaks had a higher theoretical xylem conductivity due to their higher vessel density compared to the current oaks growing under much warmer conditions. In Harbin, the larger vessel frequencies of urban trees compensated for the decrease in vessel dimensions, so the theoretical xylem conductivity was always higher than that of rural trees [84].

### 4.2. Effects of Air Pollution on the Anatomical Characteristics of Trunk Xylem

Many studies have shown that environmental pollution will have a negative impact on tree growth [5,45]. PM and Al, Ba, and Zn in the air have significant effects on the interannual change of urban tree growth [5]. Moser-Reischl et al. also found a decline in *Khaya senegalensis* growth in downtown Hanoi compared to rural trees [28]. This may be due to low groundwater levels and high pollution. Among the two pollution concentration gradients in Harbin, CN, CTA, CWT, Kh, and TB2, all decreased significantly in the heavily polluted area (Figure 7). Among these anatomical parameters, the differences in CWT and TB2 were greater than those of urban and rural controls (Figure 7C,D). These differences were mainly caused by differences in pollution because the climate of the two sampling sites is relatively close. For *P. sylvestris* var. *mongolica*, pollution caused a strong decrease in CWT and TB2 parameters related to cell wall formation, but urban heat islands did not induce such changes. Pollution may affect the physiological activities of urban trees, leading to a decline in tree growth. Sensuła et al. (2015) found that the pollution emissions would cause serious disturbances to tree metabolism, resulting in a sharp decline in tree growth. Despite the apparent long-term decline in tree growth caused by pollution, tree growth will resume after the pollution slows [44]. Air pollution and drought in the 1950s destabilized pine populations and caused more severe damage to the trees [42]. Conversely, environmental pollution also appears to have a positive effect on tree growth. [34]. In Harbin, air pollution reduced the vessel number of trees (Figure 6A and Figure 7A). However, distinctly different results were published. Rao et al. found that the characteristics of xylem of trees growing under pollution stress had a relatively large number of vessels [87]. Coal-smoke pollutants led to an increase in the number of vessels with reduced sizes of *Euphorbia hirta* L. [88]. This may be because the large amount of dust pollution in Harbin hinders the photosynthesis and transpiration of trees, thus slowing down the formation of vessels and causing their numbers to decline.

The vessel density results in Harbin were more consistent with the conclusions of most studies. That is, air pollution increased the vessel density of trees in heavily polluted areas (factory areas) rather than lightly polluted areas (urban forests) (Figure 6B and Figure 7B). Sukumaran found that the vessel density of *Cassia occidentalis* increased to a very significant level under the influence of air pollutants [89]. *Prosopis spicigera* Linn., growing under the influence of combined air pollutants, had a significantly higher vessel density [90]. In Brazil, the vessel density of *Cecropia glazioui* Sneth. was significantly higher in polluted sites than in unpolluted and intermediate sites [91]. However, Mahmooduzzafar et al. showed that air pollution from coal burning at the Badarpur Thermal Power Plants, New Delhi, reduced the vessel density of Indian blackberry trunks (*Syzygium cumini* Skeel.) [92]. In Harbin, the increasing vessel density of wood under the stress of polluted environments resulted in a lower vulnerability of polluted trees [93].

Air pollution in Harbin also reduced the vessel area (Figure 6C and Figure 7E). Although the cumulative area of all vessels (CTA) decreased, the mean percentage of vessels (RCTA) in the xylem of broadleaf trees increased (Figure 6D), which corresponds to a reduction in the increment of tree growth (Figure 4). Mahmooduzzafar et al. found that the vessel area on transverse surfaces of Indian blackberries was lower than that in unpolluted areas [92]. However, Sukumaran showed that the vessel area of *Cassia occidentalis* significantly increased under the influence of air pollutants [89]. In addition, coal smoke caused an increase in the percentage of wood vessels of *Prosopis cineraria* in Delhi, India [12]. In Harbin, air pollution inhibited pigment concentration, NR activity, and sugar content, and it promoted stomatal index and nitrate and sulfur contents and resulted in low stomatal conductance, leading to a decrease in net photosynthetic rate and, thus, a reduction in the vessel area [12].

### 4.3. Uncertainty in Interpretation of Results of Urban and Rural Growth and Xylem Anatomical Differences

Our research did not take into account the differences between Harbin and its surrounding rural in soil, elevation, natural forest, and plantation forest, which would lead to differences in the growth and anatomical characteristics of trees between urban and rural areas. *F. mandshurica* and *Q. mongolica* in the rural areas (Maoershan) are natural forests with an average age of more than 60 years, while those in the urban area (Harbin) are planted forests with an average age of 50 to 60 years. Natural forests may grow slower than plantations due to competition. However, trees in natural forests are more resistant to pollution. In addition, there is a difference in altitude between Maoershan and Harbin. Trees grow higher in the rural area (Maoershan) than in the urban area (Harbin). As the altitude rises, the temperature will drop, which will reduce the growth of trees. However, the lower altitude of Harbin will cause the temperature to rise. At the same time, lower altitudes can worsen droughts and may also affect growth. Furthermore, the rural area (Maoershan) is a natural coniferous forest soil with rich nutrients, while the urban area (Harbin) is an unnatural soil after human disturbance, and the soil fertility and texture are worse than those in rural. Therefore, from the perspective of soil condition, the quality of urban soil will be much lower than that of rural soil, which may have negative effects. We ignored these differences for now, but the impact of these differences is still significant.

In addition, there are still many deficiencies in this study. The Harbin research areas and rural areas are 87 km apart, but the two sites in Harbin are only 14 km apart, and the detailed climate data of these two sites have not been obtained yet. Currently, the pollution situation of the two sampling sites can only be inferred from the surrounding environment. Pollution data for rural areas (Shangzhi City) are also too scant. The research method in this study is not deep enough, and many contents remain unexplored. Future research in this direction can be started from the differences of physiological processes and chemical composition of trees caused by urbanization.

## 5. Conclusions

There were significant differences in radial growth and xylem anatomical characteristics between urban and rural populations in both broadleaf trees and conifers. On parameters such as VN, CTA, and Kh, trees in rural areas were higher than those in urban areas. However, broadleaf trees in urban areas generally had higher VD, RCTA, and Ks, which was not observed in conifers. Meanwhile, the overall mean thickness of all cell walls (CWT) and the overall mean cell-wall reinforcement index (TB2) of *P. sylvestris* var. *mongolica* were strongly reduced in heavily polluted areas. This may mean that conifers are more sensitive to urbanization because they do not have the mechanisms to counteract the decrease in the vessel number, area, and hydraulic conductivity caused by the urban heat island effect and air pollution, and they are strongly affected in cell-wall formation. Among the two different broadleaf tree species, *F. mandshurica* had almost no significant correlation with climatic factors, and it may be more adaptable to the urban environment than *Q. mongolica*. Warming and drying in Harbin may be an important factor affecting tree growth.

## Figures and Tables

**Figure 1 biology-12-01373-f001:**
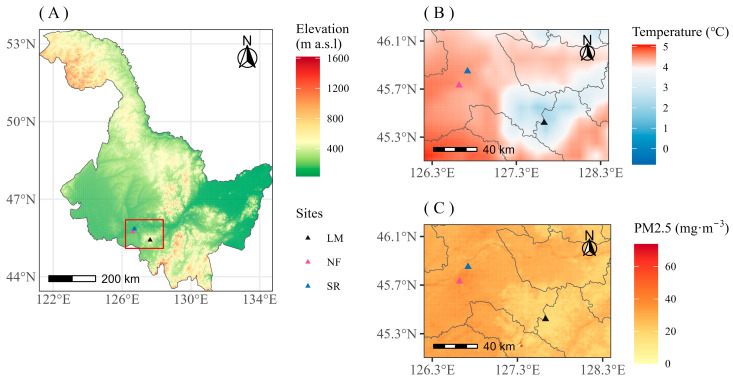
Three sample sites distributed in Heilongjiang province, Northeast China (**A**). NF (slightly polluted urban site)–Northeast Forestry University Urban Forestry Demonstration Base, Harbin, northeast China; SR (heavily polluted urban site)–Shuini Road, Harbin, Northeast China; LM (rural site)–Maoershan Forest Ecosystem National Field Scientific Observation and Research Station, Shangzhi, Northeast China. (**B**) Temperature differences among the three sampling sites (2015); (**C**) PM_2.5_ concentration differences among the three sampling sites (2001).

**Figure 2 biology-12-01373-f002:**
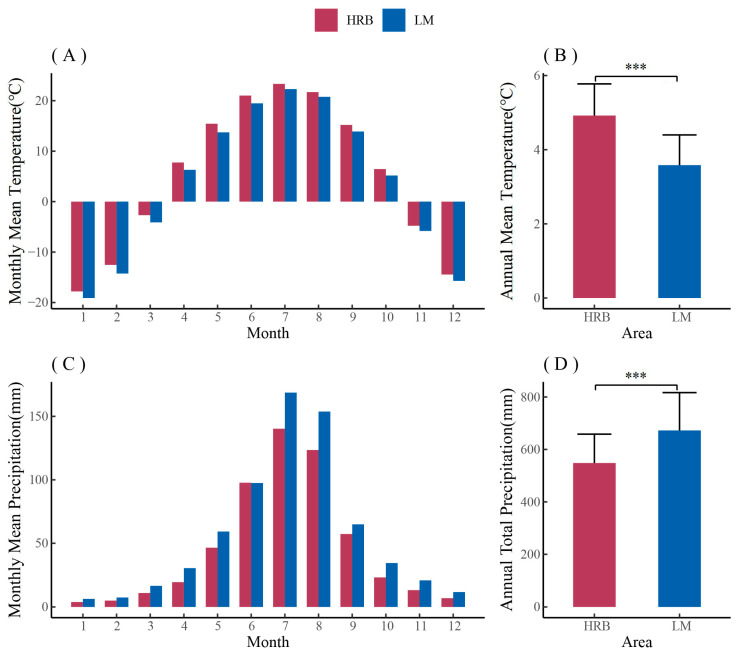
Comparisons of monthly mean temperature (**A**), annual mean temperature (**B**), monthly mean precipitation (**C**), and annual total precipitation (**D**) between urban (HRB) and rural (LM) (1951–2020). Standard deviation was shown with error bars. The significance values of the annual climate data are indicated. *** *p* < 0.001.

**Figure 3 biology-12-01373-f003:**
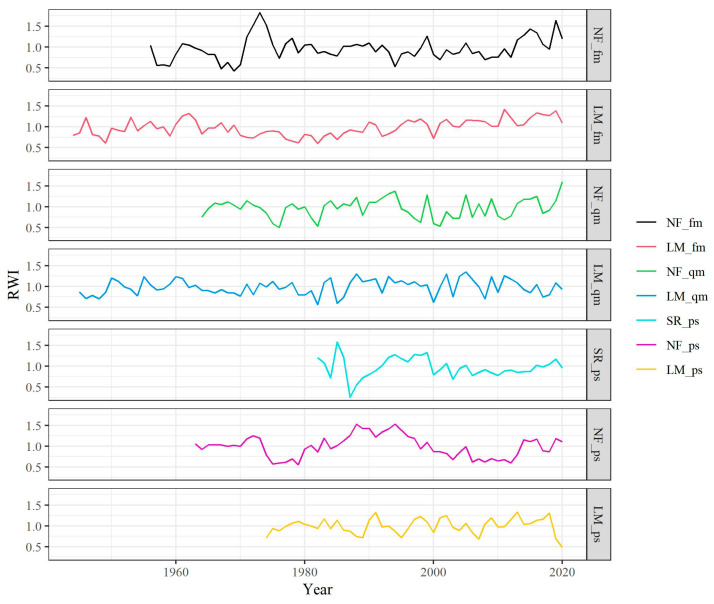
Tree-ring chronologies of *F. mandshurica* (fm), *Q. mongolica* (qm) and *P. sylvestris* var. *mongolica* (ps) xylem among urban (SR—heavy pollution, NF—light pollution) and rural (LM) sites.

**Figure 4 biology-12-01373-f004:**
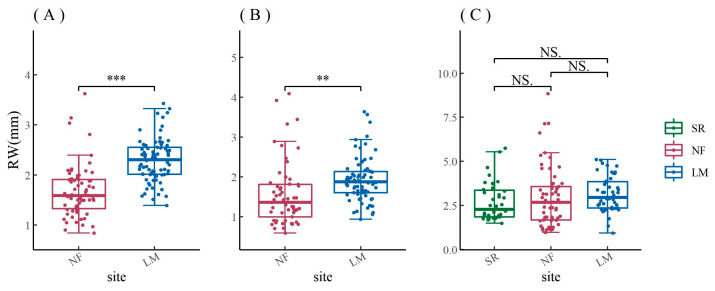
Comparison of the ring width of *F. mandshurica* (**A**), *Q. mongolica* (**B**), and *P. sylvestris* var. *mongolica* (**C**) among urban (SR—heavy pollution, NF—light pollution) and rural (LM) sites. The significance values of the ring width are indicated. NS. no significant difference; ** *p* < 0.01; *** *p* < 0.001.

**Figure 5 biology-12-01373-f005:**
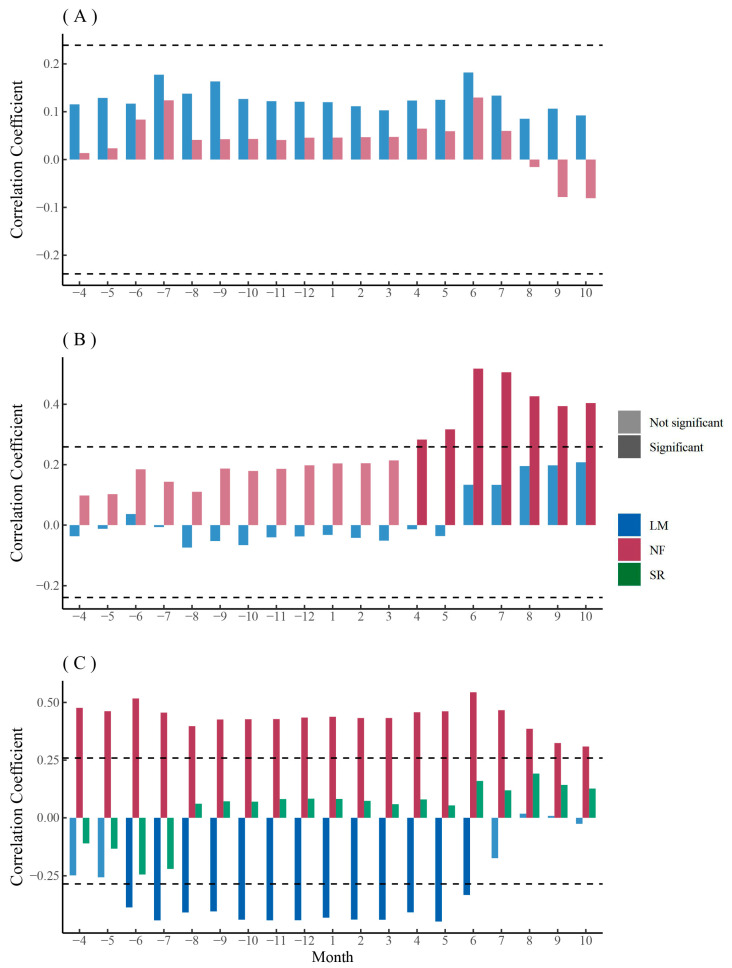
Correlation analysis between the tree-ring chronologies and monthly SPEI of *F. mandshurica* (**A**), *Q. mongolica* (**B**), and *P. sylvestris* var. *mongolica* (**C**) among urban (SR—heavy pollution, NF—light pollution) and rural (LM) sites. The dash line indicates the 0.05 significant level.

**Figure 6 biology-12-01373-f006:**
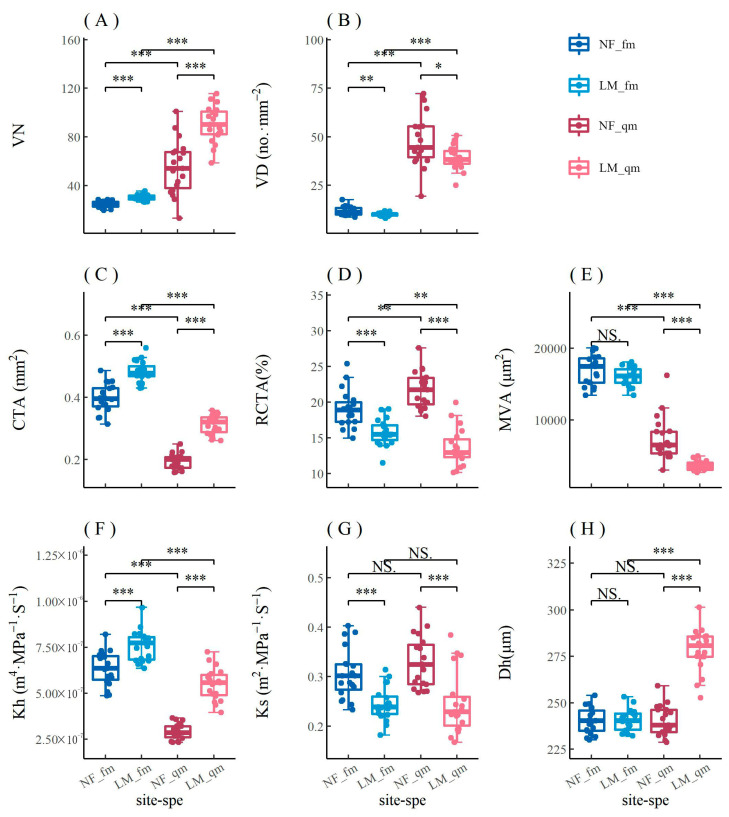
Comparison of mean vessel number (VN, **A**), vessel density (VD, **B**), cumulative area of all counted vessels (CTA, **C**), mean percentage of conductive area within xylem (RCTA, **D**), mean vessel area per rings (MVA, **E**), theoretical hydraulic conductivity (Kh, **F**), theoretical xylem-specific hydraulic conductivity (Ks, **G**), and overall mean hydraulic diameter (Dh, **H**) of the xylem of *F. mandshurica* (fm) and *Q. mongolica* (qm) between urban (NF) and rural (LM) areas. The significance values of the xylem anatomical characteristics are indicated. NS. no significant difference; * *p* < 0.05; ** *p* < 0.01; *** *p* < 0.001.

**Figure 7 biology-12-01373-f007:**
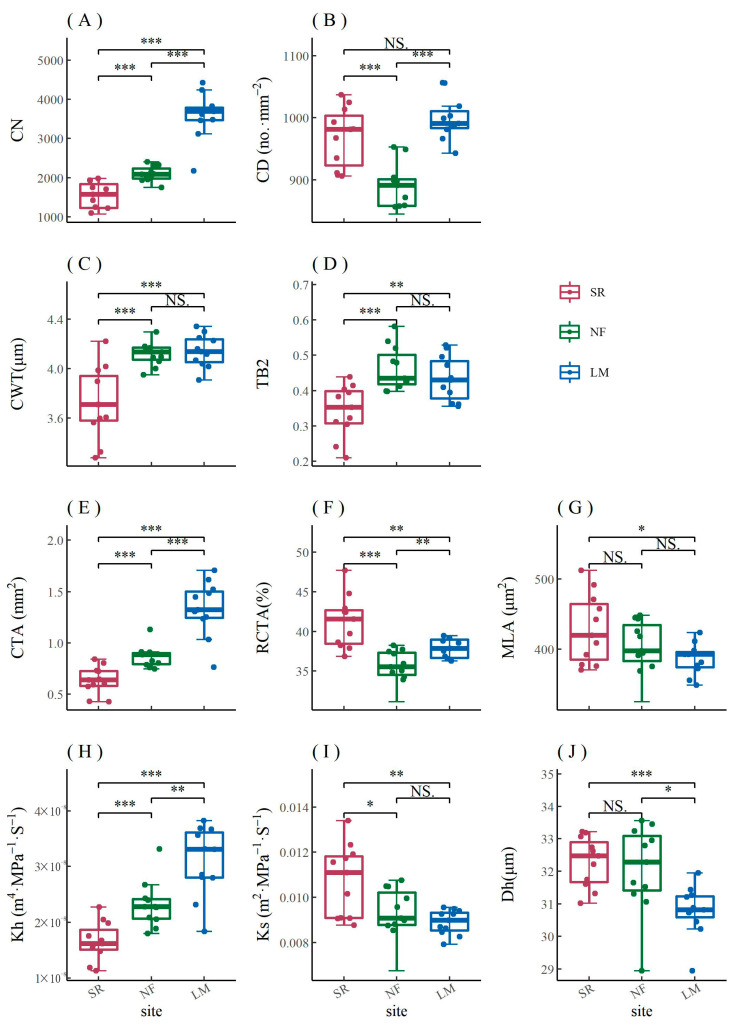
Comparison of cell number (CN, **A**), cell density (CD, **B**), cell-wall thickness (CWT, **C**), cell-wall reinforcement index (TB2, **D**), cumulative total cell area (CTA, **E**), mean percentage of conductive area within xylem (RCTA, **F**), mean lumen area (MLA, **G**), theoretical hydraulic conductivity (Kh, **H**), theoretical xylem-specific hydraulic conductivity (Ks, **I**), and mean hydraulic diameter (Dh, **J**) of *P. sylvestris* var. *mongolica* xylem among urban (SR—heavy pollution, NF—light pollution) and rural (LM) sites. The significance values of the xylem anatomical characteristics are indicated. NS. no significant difference; * *p* < 0.05; ** *p* < 0.01; *** *p* < 0.001.

**Figure 8 biology-12-01373-f008:**
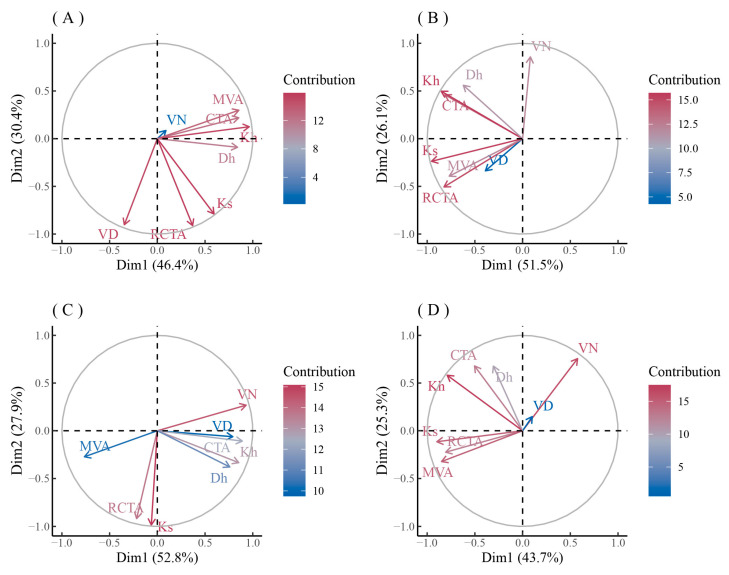
PCA analysis of xylem anatomical characteristics of *F. mandshurica* (**A**,**B**) and *Q. mongolica* (**C**,**D**) in urban (**A**,**C**) and rural (**B**,**D**) areas. Different colors represent the contribution of different factors.

**Figure 9 biology-12-01373-f009:**
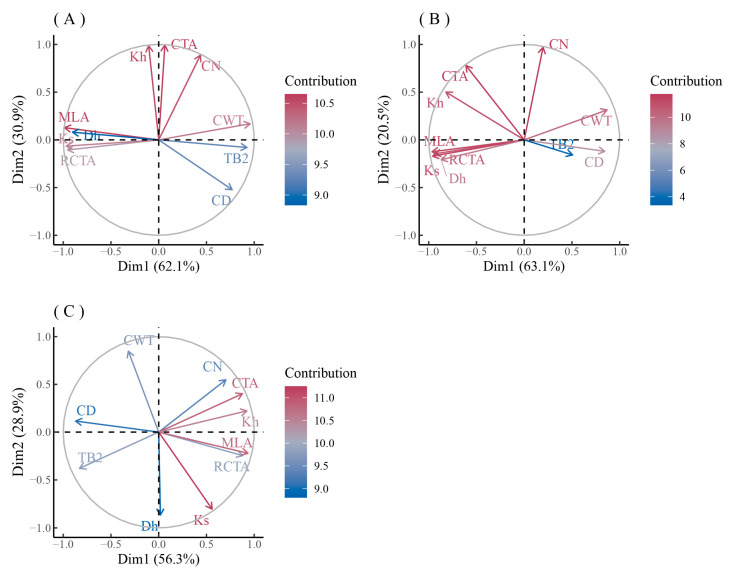
PCA analysis of xylem anatomical characteristics of *P. sylvestris* var. *mongolica* in urban (**A**—heavy pollution, **B**—light pollution) and rural (**C**) sites. Different colors represent the contribution of different factors.

**Figure 10 biology-12-01373-f010:**
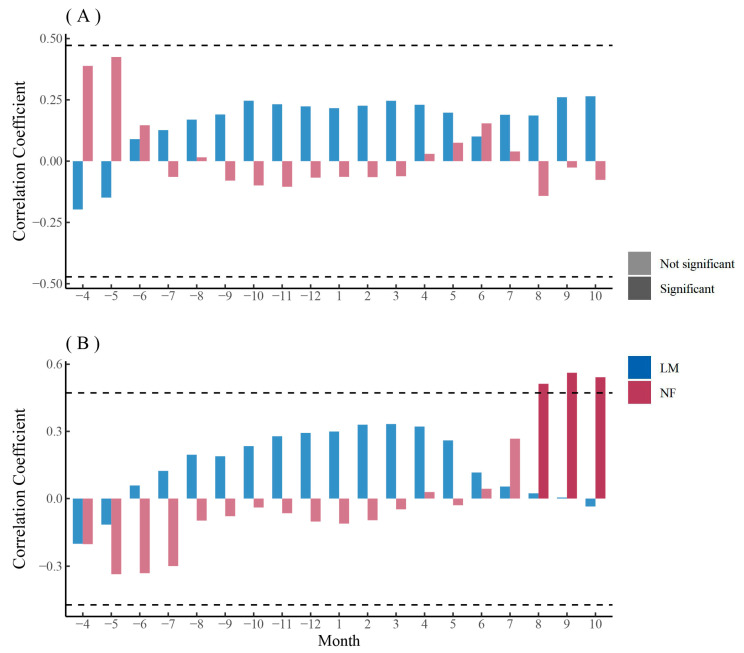
Correlation analysis between the first (**A**) and second (**B**) principal components of xylem anatomical parameters and monthly SPEI of *F. mandshurica* in urban (NF) and rural (LM) areas. The dash line indicates the 0.05 significant level.

**Figure 11 biology-12-01373-f011:**
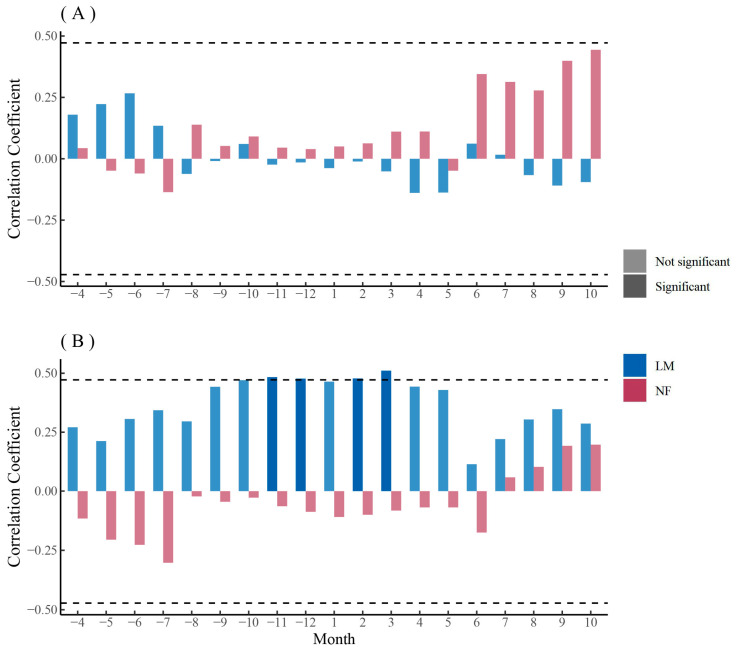
Correlation analysis between the first (**A**) and second (**B**) principal components of xylem anatomical parameters and monthly SPEI of *Q. mongolica* in urban (NF) and rural (LM) areas. The dash line indicates the 0.05 significant level.

**Figure 12 biology-12-01373-f012:**
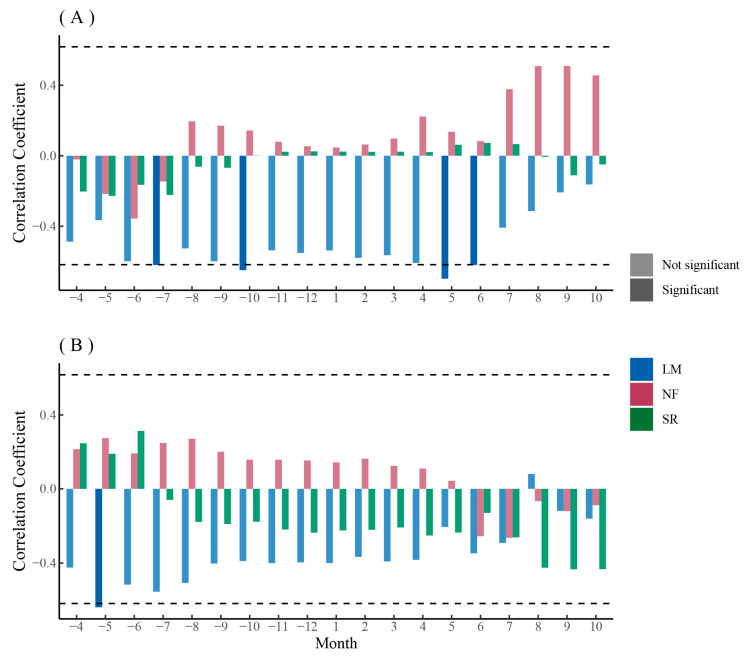
Correlation analysis between the first (**A**) and second (**B**) principal components of xylem anatomical parameters and monthly SPEI of *P. sylvestris* var. *mongolica* among urban (SR—heavy pollution, NF—light pollution) and rural (LM) sites. The dash line indicates the 0.05 significant level.

**Table 1 biology-12-01373-t001:** Sampling sites, tree species, and sample information.

Site		Species	No. of Trees	No. of Cores	Mean DBH	Mean Age
Urban	NF	*F. mandshurica*	15	30	25.440	59
		*Q. mongolica*	14	29	23.529	51
		*P. sylvestris* var. *mongolica*	15	30	31.853	51
	SR	*P. sylvestris* var. *mongolica*	15	30	22.593	35
Rural	LM	*F. mandshurica*	15	30	35.647	62
		*Q. mongolica*	13	26	34.087	69
		*P. sylvestris* var. *mongolica*	15	30	35.040	43

**Table 2 biology-12-01373-t002:** The length, standard deviation (SD), mean sensitivity (MS), first-order autocorrelation (AC1), mean inter-series correlation (Rbar), signal-to-noise ratio (SNR), expressed population signal statistic (EPS), and average growth rate (AGR) of the tree-ring chronologies.

	Length (Year)	SD	MS	AC1	Rbar	SNR	EPS	AGR
NF_fm	65 (1956–2020)	0.276	0.297	0.580	0.467	13.121	0.929	1.690
LM_fm	77 (1944–2020)	0.195	0.205	0.577	0.424	11.031	0.917	2.252
NF_qm	57 (1964–2020)	0.235	0.357	0.233	0.505	14.295	0.935	1.443
LM_qm	76 (1945–2020)	0.184	0.262	0.136	0.597	19.247	0.951	1.869
SR_ps	39 (1982–2020)	0.241	0.352	0.328	0.189	3.264	0.765	2.579
NF_ps	58 (1963–2020)	0.261	0.226	0.786	0.529	16.831	0.944	2.542
LM_ps	47 (1974–2020)	0.186	0.261	0.269	0.574	20.21	0.953	3.022

**Table 3 biology-12-01373-t003:** Two-way ANOVA analysis of the effects of site (urban and rural) and species (*F. mandshurica* and *Q. mongolica*) on xylem anatomical parameters.

		Df	Sum Sq	Mean Sq	*F*-Value	*p*-Value
VD	Site	1	592	592	10.09	0.002
	Species	1	20,652	20,652	351.62	<0.001
	Site × species	1	291	291	4.96	0.029
VN	Site	1	8044	8044	44.61	<0.001
	Species	1	39,138	39,138	217.06	<0.001
	Site × species	1	4256	4256	23.6	<0.001
CTA	Site	1	0.197	0.197	163.12	<0.001
	Species	1	0.669	0.669	554.68	<0.001
	Site × species	1	0.005	0.005	4.09	0.047
Dh	Site	1	7005	7005	100.14	<0.001
	Species	1	6794	6794	97.13	<0.001
	Site × species	1	6819	6819	97.48	<0.001
Kh	Site	1	6.88 × 10^−13^	6.88 × 10^−13^	110.09	<0.001
	Species	1	1.46 × 10^−12^	1.46 × 10^−12^	233.83	<0.001
	Site × species	1	8.41 × 10^−14^	8.41× 10^−14^	13.46	<0.001
Ks	Site	1	0.098	0.098	38.01	<0.001
	Species	1	0.003	0.003	1.05	0.309
	Site × species	1	0.002	0.002	0.89	0.349
MVA	Site	1	1.03 × 10^8^	1.03 × 10^8^	26.34	<0.001
	Species	1	2.30 × 10^9^	2.30 × 10^9^	588.80	<0.001
	Site × species	1	3.77 × 10^7^	3.77 × 10^7^	9.65	0.003
RCTA	Site	1	610.7	610.7	103.96	<0.001
	Species	1	1.5	1.5	0.26	0.614
	Site × species	1	111.0	111.0	18.89	<0.001

## Data Availability

Datasets are available on request. The raw data supporting the conclusions of this article will be made available by the authors, without undue reservation.

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
