# Peer review of "The Impact of Urbanization on Tree Growth and Xylem Anatomical Characteristics"

_biology, 2023, doi:10.3390/biology12111373_

Round 1
Reviewer 1 Report
The Authors undertook and interesting study of the effects of urban heat island and air pollution on the forest tree growth on example of three forest sites located at and near Harbin urban area. I think the research is well design and worth to be published.
The authors concluded that both the heat island with higher VPD and air pollution caused the similar effects slowing the tree ring growth, decreasing the xylem vessel area, and increasing the xylem vessel density. However, they also showed the increased hydraulic conductivity in urban trees. Could it be explained in more details, because it seems that the increase in hydraulic conductivity would be unnecessary for trees to develop if it would be assumed (and authors assumed it) that the urban trees are under the urban stress conditions with the slowing growth. Or maybe this the secondary unexpected effect of the increased vessel density?
Sincerely yours,
Author Response
Thank you very much for giving us a chance to revise our manuscript (2650952). We greatly appreciate the reviewers for their valuable and constructive comments. We took these comments into full consideration when revising the manuscript.
In the results, the hydraulic conductivity in rural areas is higher, which is the result of more vessels and larger vessel area. However, in terms of unit area, urban trees have higher vessel density and area percentage, and therefore higher xylem-specific hydraulic conductivity. On the one hand, this can be used to demonstrate a reduction in tree width growth. On the other hand, it has been confirmed that trees sought a trade-off between maximizing conductivity and creating safety margins against cavitation and embolism, that is, resulting in smaller vessel area and greater vessel density in order to adapt to a drier, warmer environment (Islam et al., 2018).
We hope our revision and responses can answer the reviewers’ questions and effectively raise the manuscript to the level of this journal. Thank you for all your help in processing and reviewing our manuscript.
Reviewer 2 Report
The article titled “The impact of urbanization on tree growth and xylem anatomical characteristics”, deals with the urban influence on the growth of trees, both planifolia and coniferous, as a result of polluting gases and heat emissions emitted by populations. This influence affects urban vegetation, creating stress situations for urban trees. This study is contrasted with the growth of trees in rural areas, for this a root study and the characteristics of the xylem are carried out. The authors claim to ignore the differences in soil and altitude between natural forests and plantations. These factors are important in tree growth, and not taking these factors into account can lead to errors.
It is an interesting study that highlights the importance of urban influence on nature. It is a very interesting investigation with results and discussion in accordance with objectives and methodology. However, and with the aim of improving the study, due to not taking into consideration certain factors such as: soil, slope, orientation, altitude, etc., we advise the authors to include a table with the species studied in both environments, rural and urban, with the parameters previously mentioned.
Author Response
Thank you very much for giving us a chance to revise our manuscript (2650952). We greatly appreciate the reviewers for their valuable and constructive comments. We took these comments into full consideration when revising the manuscript.
As suggested, we plan to add a table to the appendix as follows:
Table A1. Sampling sites information.
|
Site |
Species |
Forest |
Slope |
Orientation |
Altitude |
|
|
Urban |
NF |
F. mandshurica |
Artificial pure forest |
Level ground |
141m |
|
|
Q. mongolica |
141m |
|||||
|
P. sylvestris var. mongolica |
141m |
|||||
|
SR |
P. sylvestris var. mongolica |
132-140m |
||||
|
Rural |
LM |
F. mandshurica |
Natural secondary forest |
Gully |
Dark slope |
383m |
|
Q. mongolica |
Uphill |
Sunny slope |
395-420m |
|||
|
P. sylvestris var. mongolica |
Middle and downhill |
Sunny slope |
361m |
|||
For soils, no distinction was made in the table as all the sampling sites had dark brown soils.
We hope our revision and responses can answer the reviewers’ questions and effectively raise the manuscript to the level of this journal. Thank you for all your help in processing and reviewing our manuscript.